# Ellipsometric Study on the Uniformity of Al:ZnO Thin Films Deposited Using DC Sputtering at Room Temperature over Large Areas

**DOI:** 10.3390/ma16206644

**Published:** 2023-10-11

**Authors:** Cecilia Guillén, Juan Francisco Trigo

**Affiliations:** Centro de Investigaciones Energéticas, Medioambientales y Tecnológicas (CIEMAT), Avda. Complutense 40, 28040 Madrid, Spain; juanfrancisco.trigo@ciemat.es

**Keywords:** metal oxides, thin films, optical transmittance, electrical resistance

## Abstract

Al-doped ZnO combines high transparency and conductivity with abundant and non-toxic elements; making it suitable for optoelectronic devices with large-scale applications. In order to check the quality of the material deposited over large areas, spectroscopic ellipsometry is a powerful technique that allows the determination of various optical and electrical parameters by applying suitable oscillator models. This technique is used here to obtain sheet resistance and visible transmittance data at several equidistant points of Al:ZnO thin films deposited using DC sputtering on 15 cm × 15 cm glass substrates. Independent measurements using other optical (spectrophotometry) and electrical (four point probe) methods show analogous visible transmittance but somewhat higher resistance values than those obtained with ellipsometry, which is explained by the contribution of grain-boundary scattering compared to in-grain properties provided using ellipsometry. However, the mapping of the data gives a similar spatial distribution to the different types of measurement; therefore, proving the capacity of ellipsometry to study with a single tool the uniformity of the optical and electrical characteristics of large areas.

## 1. Introduction

ZnO is a transparent conductive oxide (TCO), showing high optical transmittance in the visible range and low electrical resistivity in its native structure [1]. The increment in n-type conductivity is relatively easy to achieve using excess Zn or doping with group-III elements (Al, Ga, and In) as Zn substituents [2]. Aluminum doping has a particular interest, because it results in a highly conductive material (Al:ZnO or AZO) constituted of abundant and non-toxic elements, making a suitable TCO alternative for diverse optoelectronic devices with large-scale deployment, such as in tunable color filters [3], smart windows [4] and photovoltaic solar cells [5]. Typically, the above applications demand a visible transmittance of T_V_ = 80–90% and a sheet resistance of *R_s_* = 10–30 Ω/sq [6], with a figure of merit defined as [7] φ = T_V_^10^/*R_s_*, which is used to compare different TCOs.

AZO thin films have been prepared using various chemical and physical techniques: electrodeposition [2], sol-gel [8], spray pyrolysis [9], evaporation [10], sputtering [11], etc. More concretely, direct current (DC) magnetron sputtering can produce transparent and conductive AZO layers at room temperature [11] on heat-sensitive substrates [12], while most of the other techniques require high substrate temperature [9,10] or thermal post-treatment above 300 °C [2,8] to achieve a good-quality material. Previous works have shown the influence of process parameters on the characteristics of AZO thin films prepared using DC sputtering on unheated glass substrates [13,14].

Another important technical challenge is to achieve high uniformity of the AZO characteristics over a large area. Typical characterization requires profilometry to determine surface roughness and film thickness; spectrophotometry to determine optical transmittance; and four point probe (FPP) electrical measurements to obtain sheet resistance [13]. Spectroscopic ellipsometry is another characterization technique that has been used to map the film thickness on different substrate areas [11,15], also allowing the simultaneous determination of several optical and electrical parameters by applying suitable oscillator models [16].

In this work, AZO thin films have been deposited using DC sputtering on unheated glass with a 15 cm × 15 cm area, and their uniformity has been analyzed with variable-angle spectroscopic ellipsometry performed on several points throughout the glass substrate. The optical constants (n, k) and electrical parameters (free carrier concentration, mobility) have been accurately obtained using a combination of the Drude oscillator model and Bruggeman effective medium approximations, including the simulated substrate and a rough top layer in the optical model, as reported by other authors [4,15]. The ellipsometric data maps are in good agreement with the analogous maps, made with visible transmittances determined using spectrophotometry and sheet resistances given by FPP electrical measurements. Therefore, we show the potential of using ellipsometry and proposed optical models to map the optical and electrical characteristics (and thus the figure of merit) of AZO coatings for large-area applications.

## 2. Materials and Methods

AZO thin films were prepared on 15 cm × 15 cm × 2 mm soda lime glasses (SLG) with DC magnetron sputtering at room temperature, using a homemade vacuum deposition system. The substrate was placed in a vertical stainless steel frame and moved in front of a rectangular target (45 cm high, 13 cm wide, 6 mm thick) consisting of 98 wt% ZnO and 2 wt% Al_2_O_3_. After the chamber was evacuated below 1 × 10^−4^ Pa, high purity Ar and O_2_ were introduced until they reached a process pressure of 4 × 10^−1^ Pa. Then, a DC power was applied to the target (set to 1.7 W/cm^2^) for 7 min to obtain a film thickness of 0.80 ± 0.04 μm, as was measured after deposition with a Dektak 3030 profilometer (Veeco, Herzogenrath, Germany). These preparation parameters were selected according to previous works [12,13], in order to obtain AZO layers with the desired visible transmittance (T_V_~85%) and sheet resistance (*R_s_*~20 Ω/sq).

Optical transmittance was measured with unpolarized light at normal incidence, using the spectrophotometer Avantes AS-5216 (Avantes, Apeldoorn, The Netherlands). Sheet resistance was obtained with a Signatone FPP head, combined with a Keithley 2450 (Keithley Instruments, Germering, Germany) Source-Measurement Unit. These optical and electrical data are compared with the information extracted from a Semilab SE-2000 Spectrocopic Ellipsometer [17] (Semilab Inc., Prielle Kornéli, Hungary). To ensure uniformity, all the measurements were performed at several points placed at 3 cm from each other on the 15 cm × 15 cm sample area.

The ellipsometric parameters (Ψ, Δ), defined as the ratio of the reflection coefficients for p- and s-polarizations [16], *R_p_*/*R_s_* = tan(Ψ)exp(*i*Δ), were acquired in the wavelength range λ = 300–2100 nm at three incident angles Φ_0_ = 55, 60 and 65°. In order to extract useful information, the structure of the sample and the corresponding optical dispersions must be modeled, whilst taking into account that the ellipsometric model must distinguish the glass substrate from the AZO features, and AZO depth heterogeneities can be considered by introducing different sublayers in the simulation [18]. The model supposes a parallel multilayer structure consisting of homogeneous, isotropic phases represented by their respective thickness *d_j_* and complex refractive index *N_j_* = *n_j_* − *ik_j_*, with *n_j_* as the real part of the refractive index and *k_j_* the extinction coefficient. During the simulation, all theoretically calculated spectra (*R_p_*/*R_s_*)*_cal_* originate from the measured ratios (*R_p_*/*R_s_*)*_meas_*, which depend on the incident angle Φ_0_, the photon energy E = hc/λ, and implicitly on the characteristics of the phases used in the model: tan(ψ)exp(iΔ) = f(Φ_0_, E, *n_j_*, *k_j_*, *d_j_*). The model parameters are thus obtained by minimizing the error function [19]:(1)$$G=\sum_{\Phi_{0}=55,60,65^{o}}{\left\{{\left(R_{p}/R_{s}\right)}_{meas}-{\left(R_{p}/R_{s}\right)}_{cal}\right\}}^{2}$$

The quality of the fit is obtained using the coefficient of determination (*r*^2^) [17], which measures the percentage of variance in the dependent variables that the independent variables explain collectively:(2)$$r^{2}=\frac{Variance\;explained\;by\;the\;model}{Total\;variance}$$

## 3. Results and Discussion

Figure 1 shows the ellipsometric parameters (Ψ, Δ) acquired at three incident angles of 55, 60 and 65° on a point of the sample. These angles gave a high *R_s_*/*R_p_* ratio, below the Brewster angle at which *R_p_* reaches zero [20]. Previously, we had measured and modeled the bare SLG substrate to obtain suitable n_SLG_ and k_SLG_ values to feed into the simulation of the SLG/AZO system. Next, the AZO characteristics were modeled, assuming there was a compact layer and rough top layer, with a combination of the Drude oscillator model (for the AZO compact layer) [15] and the Bruggeman effective medium approximation (for the rough layer, considered as a 50/50 vol% mixture of AZO and void) [18]. The coefficient of determination for this two-layer model was *r*^2^ = 91%. Subsequently, the same experimental data were simulated assuming some depth heterogeneity of the AZO characteristics, with a first compact layer close to the substrate (AZO1), a second compact layer (AZO2) and the rough top layer (AZO2 + void), which gave a better *r*^2^ = 96%. Although a somewhat higher coefficient of determination could be achieved with a four-layer model (*r*^2^ = 98%), the results do not make physical sense due to abnormally high conductivity values (*σ* > 10^6^ S/cm) at some points. Therefore, the three-layer model is considered optimum. It should be noted that, the simulated spectra start at λ = 400 nm because these models do not take into account the fundamental absorption that occurs at wavelengths below the semiconductor bandgap, which is located around 350 nm [13].

The optical parameters derived from the three-layer model are illustrated in Figure 2. It can be seen that the refractive index is above 1.7 and the extinction coefficient is below 0.02 in the visible region (λ = 400–800 nm), as reported for other AZO thin films [16,21]. Both n and k vary slowly up to λ = 1200 nm, but at larger wavelengths n decreases and correspondingly k increases sharply. The increment of k in the near infrared indicates the onset of reflection from the free carrier plasma [21], where the material enters into a metallic-like regime. The higher values of n and k obtained for AZO1, with respect to AZO2, indicate that the material initially grows with a better quality (denser and more conductive) in the region closest to the glass substrate, lowering somewhat its density and conductivity when the deposition time increases.

The Drude oscillator is used to describe the electrical conduction of free carriers in semiconductor materials [22]. From the analysis of the ellipsometric data, the dimensionless real and imaginary parts of the dielectric function corresponding to each conductive phase (*j*) are formulated as follows [3,17]:(3)$$\varepsilon_{j,1}\left(E\right)=n_{j}^{2}\left(E\right)-k_{j}^{2}\left(E\right)=-\frac{{\left(E_{j,P}/E\right)}^{2}}{1+{\left(E_{j,\Gamma }/E\right)}^{2}}$$
(4)$$\varepsilon_{j,2}\left(E\right)=2n_{j}\left(E\right)k_{j}\left(E\right)=\frac{E_{\Gamma }}{E}\;\frac{{\left(E_{j,P}/E\right)}^{2}}{1+{\left(E_{j,\Gamma }/E\right)}^{2}}$$
where *E_P_* (eV units) and *E*_Γ_ (eV units) are the plasma energy and the broadening, which is in connection with the scattering frequency.

The electrical conductivity, free carrier concentration and mobility are obtained using:(5)$$\sigma_{j}=\frac{\varepsilon_{0}E_{j,P}^{2}}{\hbar E_{j,\Gamma }}$$
(6)$$N_{j}=\frac{m_{j}^{*}\varepsilon_{0}E_{j,P}^{2}}{\hbar^{2}e^{2}}$$
(7)$$\mu_{j}=\frac{\hbar e}{m_{j}^{*}E_{j,\Gamma }}$$
where *ε*_0_ (55.263 × 10^6^ e/Vm) is the free-space permittivity, ℏ (6.582 × 10^−16^ eV s) is the reduced Planck constant, *e* is the electron charge and *m** denotes the scalar effective mass of carriers, which is assumed to be 0.25 *m*_e_ (*m*_e_ = 0.511 × 10^6^ eV/c^2^ the electron rest mass) for AZO according to the literature [23]. Furthermore, taking the simulated layer thickness for each point (*d_j_*), the sheet resistance (Ω units) is calculated as follows:(8)$$R_{j,S}=1/\left(\sigma_{j}d_{j}\right)$$

The evolution of conductivity and mobility with the carrier concentration is represented in Figure 3, which shows the data obtained for the two conductive phases (AZO1 and AZO2, in the three-layer model) at various points in the sample. Despite the considerable dispersion of values, it is observed that mobility seems to be proportional to N^−2/3^, and the conductivity proportional to N^1/3^, which corresponds to scattering by ionized impurities [24]. This is reasonable considering that optical measurements essentially provide in-grain properties, and that scattering by ionized impurities is an intrinsic property of the material. Otherwise, electrical methods include inter-grain properties measuring only the free carriers that can overcome the potential barrier at grain boundaries [23].

For each measured point in the sample, the ellipsometric fit for the three-layer model gives the thickness values that are plotted in Figure 4 as a function of the respective total thickness. The figure includes a contour map of the total thickness on the 15 cm × 15 cm sample area. The thickness variation is of 6%, in the same order as reported for other TCOs on lower substrate areas [15,18]. The map shows that the total thickness is greater at the upper and lower edges, due to a greater thickness of AZO2 and especially of the top layer (AZO2 + void), which denotes an increase in surface roughness. Such increase in thickness and roughness occurs at the edges of the sample that are attached to the substrate-holder frame. It may be due to some reflection of the sputtering plasma on the metallic frame, because the reflected particles reach the substrate with a lower energy, which reduces their diffusion during film growth [25]. Otherwise, the lateral edges of the sample remain free of the frame and show better homogeneity.

Taking the conductivity and thickness data determined by the optical dispersion model, the overall sheet resistance is calculated assuming the parallel connection of the conductive phases [18]:(9)$$R_{op}={\left(d_{1}\sigma_{1}+d_{2}\sigma_{2}+0.5d_{3}\sigma_{2}\right)}^{-1}$$

It is mapped in Figure 5, as is the sheet resistance acquired using FPP electrical measurements (*R_el_*) at the different points of the sample. Higher resistances are observed at the horizontal edges, in relation to the greater thickness of AZO2 and the top layer (AZO2 + void), which have a lower conductivity than AZO1, based on the data in Figure 3. The top layer thickness represents the surface roughness, and it is known that increased surface roughness contributes to increased resistivity [26].

At the same sample point, the values of *R_op_* and *R_el_* differ due to a discrepancy in the conductivities determined using optical or electrical measurements. Figure 6 shows these data (*σ*_op_ = *R_op_*/d_total_ and *σ*_el_ = *R_el_*/d_total_) as a function of the total film thickness. It is noted that both conductivities tend to decrease as the film thickness increases, which is related to the increase in roughness (i.e., the AZO2 + void thickness) evidenced in Figure 4. The electrical conductivity is around 650 S/cm, according to previous studies performed on AZO thin films without heating [12,13], but values above 900 S/cm are obtained optically. Such a discrepancy is usually found in the literature, due to the effect of grain boundaries. Electrical measurements give the number of free carriers that can overcome the potential barrier at the grain boundaries, but optical measurements even include the carriers with lower energy than the potential barrier at the grain boundaries. Therefore, the free carrier concentration and mobility determined optically are often somewhat higher than those given with electrical methods [22,27]. The ratios calculated in Figure 6 (*σ*_op_/*σ*_el_~1.6) are consistent with those found by other authors, indicating a similar ratio of R_DL_~1.6 [22], which is defined as the relation between the resistance to electron transport inside the lattice (1/*μ*_L_) and the resistance to electron transport due to defects such as grain boundaries and neutral impurities (1/*μ*_D_). This is because the change in conductivity is mainly related to the carrier mobility [22,28], so *σ*_op_/*σ*_el_~R_DL_ = *μ*_L_/*μ*_D_.

Regarding optical transmittance, the global spectrum at each sample point has been simulated (T_s_) [29] and compared with the corresponding spectrum measured using spectrophotometry (T_m_), as illustrated in Figure 7. For long wavelengths (λ > 1400 nm), the difference is practically zero (T_s_ = T_m_), as expected when using the Drude oscillator model, which optimally reproduces the behavior of the material in the zone of optical absorption by free carriers [22]. In the visible region (around λ = 600 nm), the simulated transmittance is somewhat lower than that obtained with spectrophotometry (T_s_ < T_m_), but in any case both increase or decrease proportionally when moving from one point to another on the sample.

For each sample point, the AZO visible transmittance (both simulated T_Vs_ and measured T_Vm_) has been calculated as the average value in the range λ = 400–800 nm from the respective SLG/AZO spectra and discounting the SLG substrate. These values are mapped in Figure 8, where it can be seen that higher visible transmittances are obtained at the upper and lower edges of the sample, related to a greater thickness of AZO2 and the top layer (AZO2 + void) in Figure 4, and corresponding also with higher resistivities in Figure 5. This shows that the increase in roughness (i.e., the AZO2 + void thickness) contributes to an increase in both electrical resistivity and visible transmittance [6].

Finally, the figure of merit defined by Haacke [7] has been calculated with the respective data obtained using the simulation (φ_s_ = T_Vs_^10^/*R_op_*) and from independent electrical and spectrophotometric measurements (φ_m_ = T_Vm_^10^/*R_el_*), which are represented in Figure 9. Although the range of variation of the visible transmittance is narrow for the simulated values (84–88%) and when directly measured (85–88%), it dominates in terms of merit. Therefore, higher quality is obtained in more transparent regions, despite the fact that their resistance is also somewhat higher. On the other hand, the simulated figure of merit is in general higher (φ_s_ > φ_m_) because the optically determined resistance is lower (*R_op_* < *R_el_*). In fact, the ratio between the maximum values φ_s_/φ_m_ = 0.022/0.014 = 1.6 is analogous to that established for the respective conductivities (*σ*_op_/*σ*_el_~1.6) in Figure 6. It should be noted that, the φ data presented here are calculated considering average values of transmittance instead of the maximum transmittance at a particular wavelength, which exceeds 90%. Even so, the figure of merit is always above 0.010 Ω^−1^, higher than that reported for other TCOs grown at high substrate temperature [30,31,32].

## 4. Conclusions

Ellipsometric data acquired for AZO thin films deposited using DC sputtering on SLG substrates show an optimal fit to a three-layer model consisting of a first compact layer that is close to the substrate (AZO1), a second compact layer (AZO2) and a rough top layer (AZO2 + void). The values obtained using optical simulation for the conductivity (*σ*_op_), free carrier concentration and mobility indicate scattering by ionized impurities, being somewhat higher than the electrical conductivity (*σ*_el_) determined using four point probe measurements. This corroborates that optical measurements provide in-grain properties, and the ratio *σ*_op_/*σ*_el_~1.6 obtained at different points of the sample gives the relation between the resistance to electron transport inside the lattice and the resistance to electron transport due to grain boundaries and neutral impurities.

The maps of sheet resistance and visible transmittance performed on the 15 cm × 15 cm sample area show the same spatial distribution whether they are obtained from ellipsometry or from independent spectrophotometry and electrical measurements. Slightly higher *R_s_* and T_V_ values are observed at the horizontal edges that are attached to the substrate-holder frame, corresponding to a greater thickness of AZO2 and of the top layer (AZO2 + void) at these edges. In any case, the variation in thickness is only 6% over the whole area and the figure of merit (T_V_^10^/*R_s_*) is always above 0.010 Ω^−1^, proving that high-quality AZO thin films are obtained using DC sputtering at room temperature on large areas.

## Figures and Tables

**Figure 1 materials-16-06644-f001:**
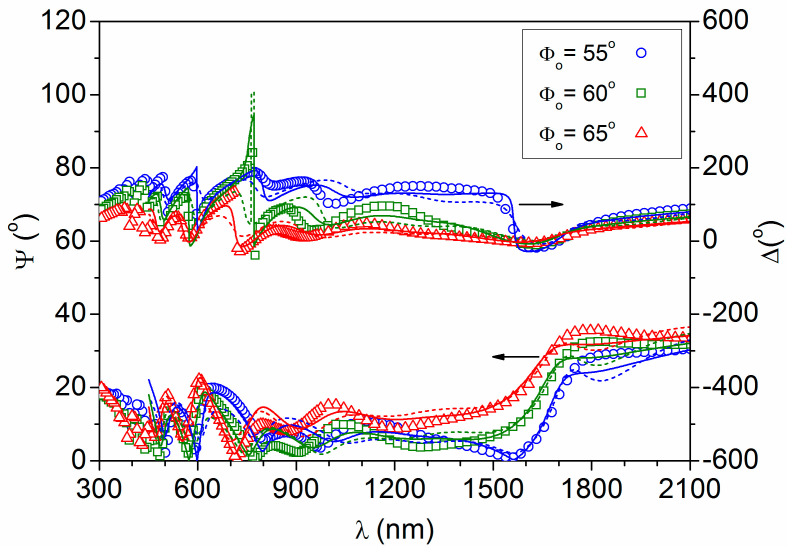
Ellipsometric data measured (symbols) and fitted (lines) for three incidence angles (Φ_0_) at one point of the SLG/AZO sample. The dashed lines correspond to the two-layer model and the solid lines to the three-layer model.

**Figure 2 materials-16-06644-f002:**
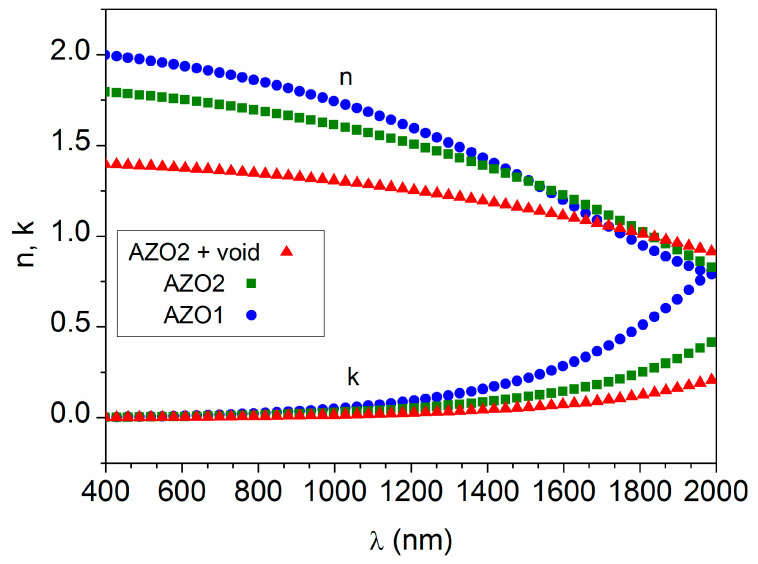
Optical parameters obtained from the data in Figure 1 simulated with the three-layer model.

**Figure 3 materials-16-06644-f003:**
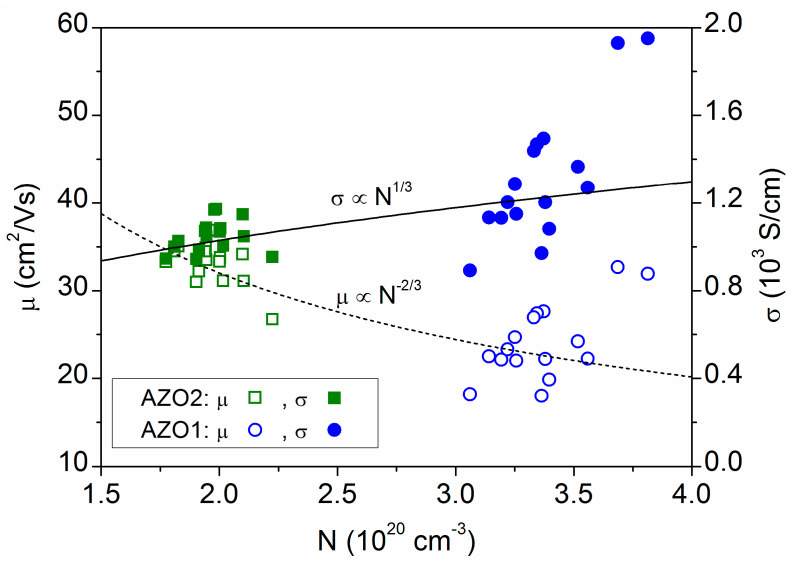
Electrical conductivity (*σ*), carrier concentration (N) and mobility (*μ*) values obtained for the two conductive phases (AZO1 and AZO2 in the three-layer model) at several equidistant points in the 15 cm × 15 cm sample area.

**Figure 4 materials-16-06644-f004:**
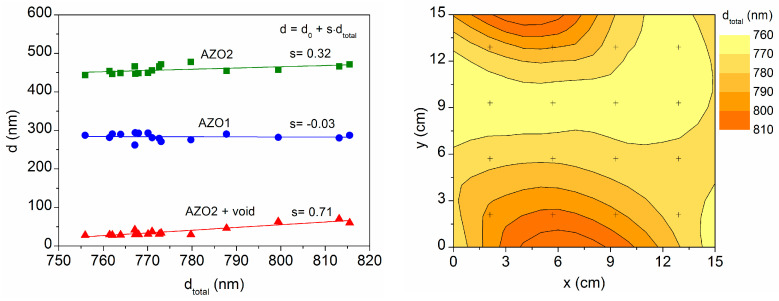
Thickness values provided using the three-layer model at sixteen equidistant points marked on the 15 cm × 15 cm sample area. The contour map represents the total thickness at each point.

**Figure 5 materials-16-06644-f005:**
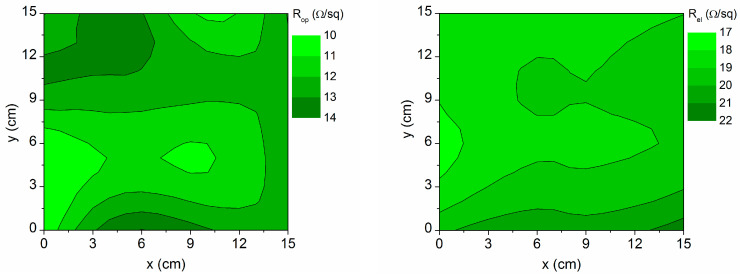
Contour map of the sheet resistance values obtained using optical simulations (*R_op_*) and electrical measurements (*R_el_*) on the 15 cm × 15 cm sample area.

**Figure 6 materials-16-06644-f006:**
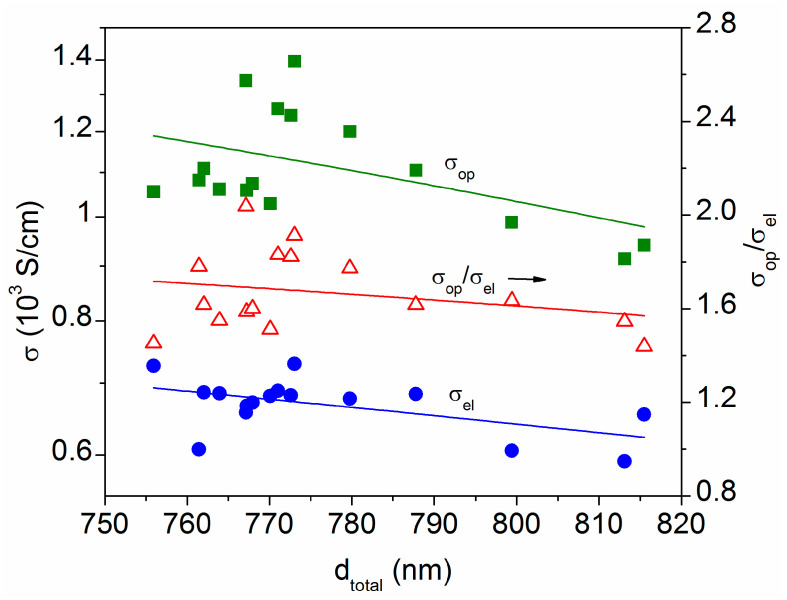
Conductivity values determined using optical simulations (*σ*_op,_ square_s_) and electrical measurements (*σ*_el_, circles) at several equidistant points in the 15 cm × 15 cm sample area, plotted as a function of the respective total thickness. The black arrow indicates the right y-axis for the ratio (triangles).

**Figure 7 materials-16-06644-f007:**
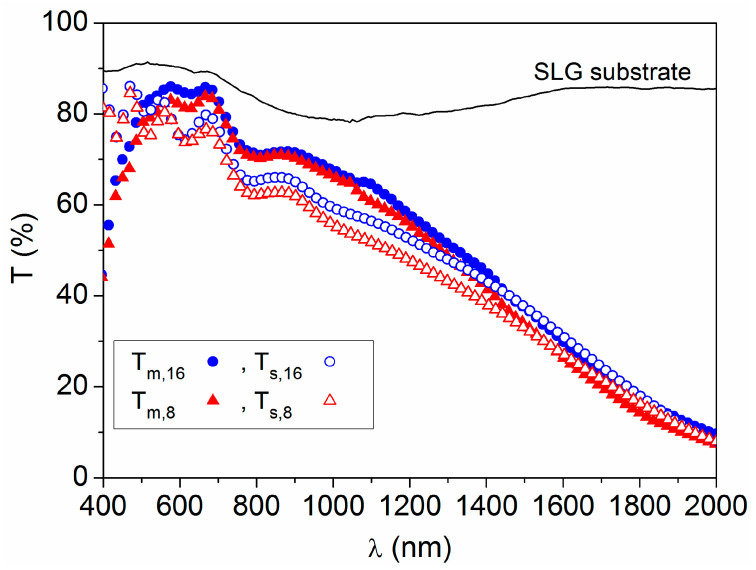
Transmittance spectra simulated using the three-layer model (T_s_) and measured with spectrophotometry (T_m_) at point 8 (x = 13, y = 6) and point 16 (x = 13, y = 13), on the SLG/AZO sample of 15 cm × 15 cm area. The measured transmittance for the bare SLG substrate is included for comparison.

**Figure 8 materials-16-06644-f008:**
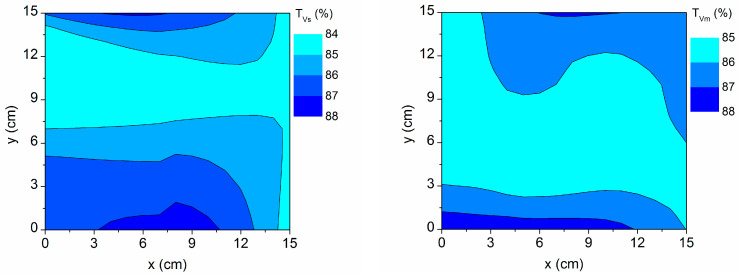
Contour map of the visible transmittance simulated with the three-layer model (T_Vs_) and measured using spectrophotometry (T_Vm_) on the 15 cm × 15 cm sample area.

**Figure 9 materials-16-06644-f009:**
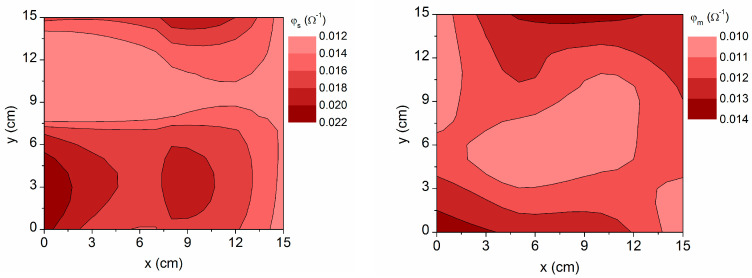
Contour map of the figure of merit obtained using simulation with the three-layer model (φ_s_) and from independent optical and electrical measurements (φ_m_).

## Data Availability

Data available on request.

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
