# Peer review of "Ellipsometric Study on the Uniformity of Al:ZnO Thin Films Deposited Using DC Sputtering at Room Temperature over Large Areas"

_materials, 2023, doi:10.3390/ma16206644_

Round 1

Reviewer 1 Report

The authors investigated the uniformity of optical and electrical properties of a large-size Al-doped ZnO thin films grown by DC sputtering and compared the ellipsometry results with other measurement methods. The topic is related to and the results would contribute to commercially fabrication of large scale homogeneous transparent conducting films based on ZnO. The investigations were well conducted, the analysis and discussion were thorough, and the report was well written. Due to my following comments, I would suggest for compulsory revision before it is considered for publication.

Comments

1. For the Al-doped ZnO thin films, they preserve the direct band gap characteristic and the band gap value is around 3.3 ~ 3.4 eV (between 300 and 400 nm in wavelength). However, in Figure 2, the band gap related peak-like characteristic in refractive index as well as the step-like increase in distinction coefficient were not observed. Is the film still Al-doped ZnO? This is a significant ambiguity needing clarification.

2. Concerning the 3-layer model, I suggest the authors to add a short description for the possible reasons leading to two conductive phase, so that the readers can better understand the physics behind it.

3. In Figure 3, the data points of both the conductivity and the mobility of AZO1 phase are scattered, and therefore the authors' claim of the N1/3 and N-2/3 dependence seems too strong. A little more reserved statement would be better.

4. In Figure 7, the subscripts "8" and "16" were not defined in both figure caption and the text, so the authors please explain them for clarity. Moreover, the difference in transmittance between Tm,16/Ts,16 and between Tm,8/Ts,8 are still significant between 1300 and 1400 nm, and therefore, I suggest to change the description in Line 220 to "For long wavelength (l > 1400 nm)......"

Reviewer 2 Report

This manuscript discusses about the ellipsometric study on the Al-doped ZnO thin films used as a transparent conductor. The manuscript is well written and it is suitable for being published in Materials.

There are only a few questions and concerns need to be addressed as following:

Major:

1.) Page 2, line 64-72: could the authors please comment on the target thickness for the deposited films and if there is any film thickness measurement by microscopy after deposition ? 

2.) Page 3, line 102: Could the authors please comment on why the incident angles of 55Ëš, 60Ëš and 65Ëš were selected here ? 

3.) Page 3, equations 3-8: could the authors please define all the parameters and give a unit to each of them ? 

4.) Page 5-6, figures 4 and 5: the authors mentioned about different points of the same were measured, could the authors please specify on this, how many points and where are the points were selected ?

Minor:

a.) Page 1, line 2, could the authors please re-format the title for the word “de-posited” ? It is okay to break a word in the main text, however, probably not a good idea for the title. 

Based on the current status of the manuscript, I recommend a minor revision.

Round 2

Reviewer 1 Report

The authors have responded to the reviewers' comments well with clear answers and reasoning in physics. The report on the optical and electrical properties of a large-area Al-doped ZnO thin films is revised accordingly, where the ambiguities are clarified. Therefore, I suggest this manuscript for publication in Materials.